**Data Availability Statement:** All relevant data are within the paper and its Supporting Information files.

**Funding:** This study based on the "Home Delivery Practice and its Predictors in South Ethiopia"

# Home delivery practice and its predictors in South Ethiopia

**Dejene Hailu**[1]*, **Henok Tadele**[2], **Birkneh Tilahun Tadesse**[3], **Akalewold Alemayehu**[1], **Teshome Abuka**[1], **Fitsum Woldegebriel**[3], **Abel Gedefaw**[4], **Selamawit Mengesha**[1], **Yusuf Haji**[1]

1 School of Public Health, Hawassa University, Hawassa, Ethiopia, 2 Department of Pediatrics and Child Health, College of Health Sciences, Addis Ababa University, Addis Ababa, Ethiopia, 3 Department of Pediatrics and Child Health, Hawassa University, Hawassa, Ethiopia, 4 Department of Obstetrics and Gynecology, Hawassa University, Hawassa, Ethiopia

☯ These authors contributed equally to this work.

* dejenkassa@yahoo.com

## Abstract

### Background

Institutional delivery is one of the key interventions to reduce maternal death. It ensures safe birth, reduces both actual and potential complications, and decreases maternal and newborn death. However, a significant proportion of deliveries in developing countries like Ethiopia are home deliveries and are not attended by skilled birth attendants. We investigated the prevalence and determinants of home delivery in three districts in Sidama administration, Southern Ethiopia.

### Methods

Between 15–29 October 2018, a cross sectional survey of 507 women who gave birth within the past 12 months was conducted using multi-stage sampling. Sociodemographic and childbirth related data were collected using structured, interviewer administered tools. Univariate and backward stepwise multivariate logistic regression models were run to assess independent predictors of home delivery.

### Results

The response rate was 97.6% (495). In the past year, 22.8% (113), 95% confidence interval (CI) (19%, 27%) gave birth at home. Rural residence, adjusted odds ratio (aOR) = 13.68 (95%CI:4.29–43.68); no maternal education, aOR = 20.73(95%CI:6.56–65.54) or completed only elementary school, aOR = 7.62(95% CI: 2.58–22.51); unknown expected date of delivery, aOR = 1.81(95% CI: 1.03–3.18); being employed women (those working for wage and self-employed), aOR = 2.79 (95%CI:1.41–5.52) and not planning place of delivery, aOR = 26.27, (95%CI: 2.59–266.89) were independently associated with place of delivery.

survey was supported by Hawassa University through its KMC Projects in the form of funds awarded to HT. The funders had no role in study design, data collection, and analysis, decision to publish, or preparation of the manuscript.

**Competing interests:** The authors have declared that no competing interest exists.

**Abbreviations:** aOR, Adjusted odds ratio; CHW, community health worker; CI, confidence interval; EDHS, Ethiopian Demographic and Health Survey; KIT, KMC implementation team; KMC, Kangaroo Mother Care; LBs, Live births; SD, Standard deviation; SNNPR, Southern Nations Nationalities and Peoples Region; SSA, sub-Saharan Africa; UNICEF, United Nations International Children's Fund; WHO, World Health Organization.

## Conclusion

The prevalence of institutional delivery in the study area has improved from the 2016 Ethiopian Demography Health Survey report of 26%. Uneducated, rural and employed women were more likely to deliver at home. Strategies should be designed to expand access to and utilization of institutional delivery services among the risky groups.

## Introduction

Maternal mortality is a major public health problem in developing countries particularly, in sub–Saharan Africa (SSA) [1]. Every year, nearly half a million women and girls needlessly die from complications of pregnancy and childbirth, and 99% of these deaths occur in developing countries [1]. According to the joint World Health Organization (WHO)/United Nations Children's Fund (UNICEF) 2015 estimate, the global maternal mortality rate was 216/100,000 live births (LBs), while 436/100,000 LBs and 546/100,000 LBs were reported for the least developed and SSA countries, respectively [2]. Based on a recent report, Ethiopia is one of the countries with the highest maternal (412/100,000), neonatal (29/1000) and infant mortalities (48/1000 LBs) [3].

Most maternal deaths which occur in developing countries are due to complications during labor and delivery, and during the immediate postpartum period. The WHO recommends that every delivery should be attended by skilled personnel. However, a significant proportion of women do not have this access during childbirth [1, 4].

Globally, it was estimated that 22% of deliveries didn't happen in health facilities by the year 2015. A similar report showed that the aggregate prevalence of home deliveries in Eastern and Southern Africa was about 38% [2]. In Ethiopia, community-based surveys conducted in various regions of the country reported 31% to 96% home deliveries [5–12], which is similar to that is reported in the Ethiopian Demographic Health Survey (EDHS) 2016 survey (73%) [3]. The consistently high prevalence of home deliveries in different parts of the country indicate that the large proportion of pregnant women and their babies are at risk of complications including death related to childbirth [1, 2].

Available evidence revealed that several factors such as socio-economic status, maternal education, infrastructure, place of residence, unpleasant experience with healthcare providers and access to primary health care services influence women's choice of place of delivery [13–16]. For example, a study conducted in Nigeria and Ethiopia showed that women in rural areas were more likely to consider facility delivery as unimportant and complained about distance and inability to pay fees [17].

According to the EDHS data–covering 2011 to 2015 –the prevalence of home birth in Southern Nations Nationalities and Peoples Region (SNNPR) was 74% [3]. However, as part of an ongoing Kangaroo Mother Care (KMC) implementation project (ClinicalTrials.gov: NCT03506698) in southern Ethiopia, very few home births were reported by the health extension workers (HEWs). While this low home birth rate could signal an improving health system in the region, it could also indicate poor tracking and reporting systems of home births by HEWs. To assess the real burden of home births in the area, we conducted a population-based survey in three districts of Sidama Administration–Hawassa City Administration, Dale and Shebedino districts. The aims of the survey were to assess the prevalence of homebirth and determinants in three districts of Sidama Administration, SNNPR.

## Methods

### Study setting

This survey was conducted at KMC implementation sites in SNNPR, which included Hawassa City Administration, Dale and Shebedino districts. Hawassa Comprehensive Specialized Hospital, Adare General Hospital, Yirgalem General Hospital, and Leku Primary Hospital served as KMC implementation centers. Hawassa is the capital city of SNNPR/Sidama Administration and it is located 275 km South of Addis Ababa, the capital of Ethiopia. Hawassa City Administration has eight sub-cities divided in 32 *kebeles* (the lowest administration unit in Ethiopia). The estimated total population size of Hawassa city in 2017 was 455,658 as projected from the 2007 Ethiopian national census [18]. It is estimated that there are over 10,000 deliveries taking place in Hawassa every year. There are 3 public hospitals and 12 health centers in the city.

Shebedino district, the second study area is located 30km South of Hawassa city and has 32 *Kebeles*. Leku town is the capital of the district. There is one primary hospital, 9 health centers and 32 health posts in the district. An estimate of 121 deliveries is attended per month at Leku Primary Hospital.

The total population of Dale district in 2017 was 317,246 with 11,104 expected deliveries per year (18). Yirgalem town is the capital of the district and it is located 45kms south of Hawassa City. There is one general hospital, 10 health centers and 36 health posts in the district.

### Study design and population

A community based cross-sectional survey was conducted during 15th- 20thOctober, 2018. Randomly selected women who gave birth in the last one year and residing at least 6 months in the area were included in the survey. Non-consenting mothers were excluded from the study.

### Sample size

The sample size was calculated using Epi info 7 Statistical software for population survey. Considering 72.5% home delivery in SNNPR (EDHS 2016), 95% confidence interval (CI), margin of error of 5%, design effect of 1.5 for a cluster of 10 and 10% non-response rate, a total of 507 women were needed [3].

### Sampling procedures

A multistage sampling technique was used to enroll study participants. There are 32 kebeles in Hawassa city, 35 in Dale and 32 in Shebedino districts. We selected 11 kebeles [4 kebeles from Hawassa City, representing urban households (36%); 4 from Dale and 3 from Shebedino districts, both representing rural households (64%)] using simple random sampling techniques. Households of women who gave birth during the last 12months preceding the study were identified and listed with the help of family folders available at the health posts of the selected 11 kebeles. Finally, the calculated sample size was proportionally allocated to the kebeles based on the identified number of eligible women. Women in each of the selected kebeles were randomly selected by simple random sampling technique using the list as a sampling frame.

### Data collection

The questionnaires were first prepared in English and then translated to local languages: "Sidamu Afoo" for rural residents and "Amharic" for urban residents. Six data collectors who completed at least first degree in public health disciplines interviewed the participants.

## Data analysis

Data analysis was done using SPSS version 25. Descriptive, bivariate, and multivariate analyses were done to assess association between sociodemographic variables and place of delivery. Odds ratios and 95% CIs were computed. A backward stepwise multivariate regression model was run using variables with *P-value* <0.2 in the bivariate analysis, which included place of residence, age, education and occupation of women, paternal education, distance of health center from home, family size, number of ANC follow up, knowing the due date, planned place of birth and birth order. Model fitness was checked using Hosmer and Lemeshow test of goodness of fit which yielded a p-value = 0;889. Level of significance for independent associations was set at p<0.05.

## Ethics approval and consent to participate

This study was approved by Institutional Review Board (IRB) of Hawassa University. Considering non-invasive nature of data collection procedures, which is a case in most surveys conducted in Ethiopia, a verbal consent, which was approved by the IRB, was obtained from all women participated in the survey after detailed introduction of the objectives of the study and the right to withdraw from the study at any time. The information sheet and consent was read slowly and loudly by the data collector to the participants. Then, they were asked if there were any queries. After the mothers had confirmed that all is clear, they were asked one last question if they were willing to participate in the survey or not. The data collectors circled either 'yes' or 'no' based on whichever is selected and the interview was conducted only if the data collector was told to circle the response 'yes'. This was attached to the questionnaire and documented.

Seven (1.4%) mothers were less than 18 years old, but the consent was obtained from these women since they have been married and do not live with the family. The IRB was aware of this situation and approved the verbal consent obtained from mothers less than 18 years old. Confidentiality was maintained by decoding study subjects' identifiers and the consent form and questionnaires were kept in locked file cabinets.

# Results

## Socio-demographic characteristics of the respondents

We interviewed 495(97.6%) mothers who had given birth 12 months preceding the survey. The mean (±SD) age of the participants was 25.85(±4.95) years; 339(68.5%) were rural residents. Two hundred and seven (42%) respondents completed grades 5–8 while 61(12.3%) did not attended any school. Majority, 377(76.2%) of the women were housewives and 45% of them had at least five family members (Table 1).

## Antenatal care follow- up and plan for place of delivery

The majority, 473(95.6%) of the women had antenatal care (ANC) follow up. Two-hundred forty-four (51.6%) women had at least 4 visits during the whole pregnancy of the index child. Three hundred sixty (64%) attended ANC in health centers. Three hundred forty-seven (70%) knew the due date of the index child. Four hundred eighty-two (97.4%) women planned their place of delivery; 309 (63.2%) preferred health centers, 102 (21%) preferred government hospitals and 62 (12.5%) preferred home delivery. The majority, 415(83.8%) of respondents wished their delivery to be attended by skilled health professionals while 80(16.2%) preferred their women relatives, or traditional birth attendants as birth attendants.

**Table 1. Socio-demographic characteristics of women with infants(0-12months) at three districts in Sidama Administration, 2018.**

| Characteristics | Frequency (%) |
|---|---|
| Age (years) | |
| ≤20 | 92(18.6) |
| 21–25 | 165(33.3) |
| 26–30 | 169(34.4) |
| >30 | 69(14) |
| Level of education | |
| No class attended | 61(12.3) |
| Grade 1–4 completed | 90(18.2) |
| Grade 5–8 completed | 207(41.8) |
| Grade 9–10 completed | 71(14.30 |
| Higher than grade 10 | 66(13.3) |
| Occupation | |
| Housewife | 381(78.1) |
| Employed | 107(21.9) |
| Paternal education | |
| No class attended | 43(8.70 |
| Grade 1–4 completed | 36(7.3) |
| Grade 5–8 completed | 164(33) |
| Grade 9–10 completed | 122(24.6) |
| Higher than grade 10 | 130(26.3) |
| Household's monthly income (in Ethiopian Birr) | |
| Low (<1000) | 170(34.3) |
| Medium (1000–1999) | 132(26.7) |
| Higher income (≥2000) | 91(18.4) |
| Unknown income(unreported) | 102(20.6) |
| Distance from house to health center | |
| <30 minutes' walk | 213(43) |
| 30–60 minutes' walk | 282(57) |
| Total Family members | |
| <5 members | 274(55.4) |
| ≥5members | 221(44.6) |

## Characteristics of deliveries

The overall prevalence of home delivery was 22.8% (113), 95%CI: 19%, 27%). Home delivery rate among women from rural areas was 32% (108), 95%CI: 27.6–36%) while it was 3.2% (5) among urban residents (Table 2). One hundred twelve (72%) of urban women gave birth at government hospitals while 166(49%) of the rural women delivered at health centers. Reasons for home delivery included inconvenience of health facilities, 44(40.7%); personal preference to deliver at home, 39(36.3%); not considered important, 20(18.5%); fear of delivering on the way to the health facility, 15 (14%); and lack of transportation, (10%) (Table 2).

The proportion of home delivery significantly varied by districts ($X^2 = 63$, DF = 2, value<0.001). In Shebedino district, majority of the women gave birth at home compared to Dale and Hawassa districts (Fig 1)

**Table 2. Delivery characteristic of participants at KMC implementation sites, Sidama Administration, 2018.**

| Variables | Categories | Frequency (%) |
|---|---|---|
| Birthplace of the index child | Health center | 193(39) |
| | Hospital | 175(35.4) |
| | Home | 113(22.8) |
| | Private clinic/hospital | 14(2.8) |
| Reasons for home births (n = 113) | Health facilities are inconvenient | 45(40) |
| | Personal preference | 41(36.3) |
| | Unnecessary to give birth at health facilities | 20(18) |
| | Fear of delivering on the way to the health facility | 16(14) |
| | No transportation | 11(40) |
| | Health Professionals (HPs) are not friendly | 3(0.6) |
| Mode of delivery of the index child | Normal vaginal delivery | 448(90.5) |
| | Caesarean Section | 47(9.5) |
| Attendant of the delivery of the index child | Doctors | 56(11.3) |
| | Nurses/Midwifery | 203(41) |
| | Unknown health professional | 132(26.7) |
| | Female relatives/friends | 57(11.5) |
| | Traditional Birth attendants | 28(5.7) |
| | No one (mother herself) | 9(1.8) |
| | Others | 25(5.1) |

## Factors associated with home delivery

Rural residence, being employed, lack of or limited education, knowing due date of the index child and planned place of delivery predicted the place of delivery.

Women residing in rural areas were 5.28 times more likely to deliver at home as compared to urban dwellers, aOR = 5.28(95%CI:1.25–22.16). Similarly, the odds of home birth among

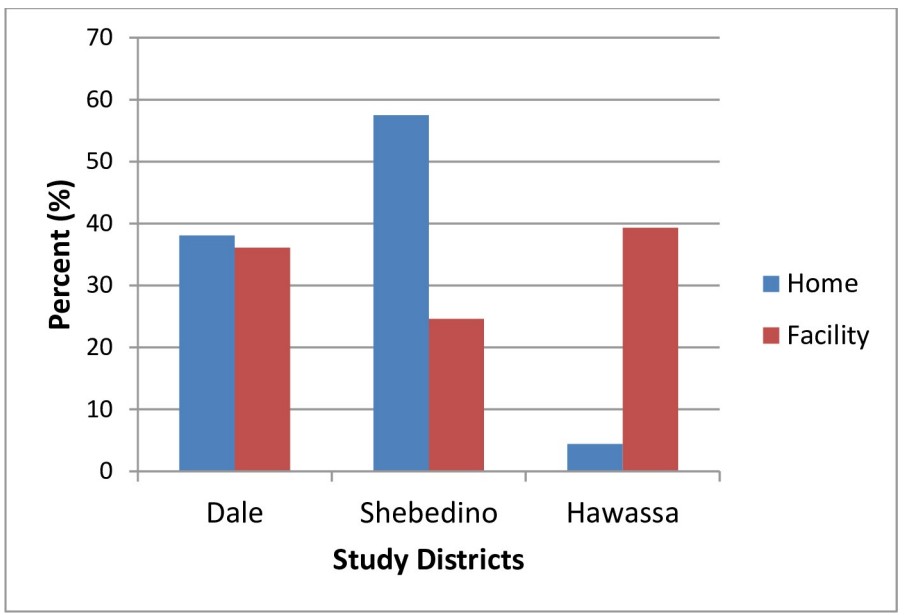

**Fig 1. Place of delivery cross tabulated by the study districts, Sidama Administration, 2018.**

uneducated 8.78(95% CI: 2.33–33.01) or minimally educated women 3.81(95% CI: 1.16–12.49) was high. The odds of home birth among women who did not know the due date of their index child was two folds higher compared to those who knew it, aOR = 2.12(95%CI: 1.21–3.71). Furthermore, women whose husbands did not go to school were about 3 times more likely to deliver at home compared to their counter parts, aOR = 3.27, 95% CI: 1.20–8.88) (Table 3).

## Discussion

This cross-sectional survey included mothers from urban and rural settings. The prevalence of home delivery was almost 23%. Rural residence, being employed and lower maternal education were among determinant factors for home birth.

In the current study, the proportion of women who gave birth at home was 22.8% (32% and 3.2% among women from rural and urban areas, respectively), a prevalence comparable to

**Table 3. Bivariate and multivariate analyses of factors associated with home deliveries at KMC implementation sites, Sidama Administration, 2018.**

| Variables | Categories | Place of delivery | | COR (95%CI) | AOR (95% CI) |
|---|---|---|---|---|---|
| | | Home (%) | Health facility (%) | | |
| Place of residence | Rural | 108(95.6) | 231(60.5) | 14.12(5.63–35.42) | **13.68(4.29–43.65)** |
| | Urban | 5(4.4) | 151(39.5) | 1.0 | 1.0 |
| Age of women | ≤20 years | 12(10.6) | 80(20.9) | 1.0 | |
| | 21–25 years | 33(29.2) | 132(34.6) | 1.67(0.81–3.41) | * |
| | 26–30 years | 46(40.7) | 123(32.2) | 2.49(1.24–4.99) | |
| | >30 years | 22(19.5) | 47(12.3) | 3.12(1.42–6.88) | |
| Maternal education | No class attended | 33(29.2) | 28(7.3) | 31.11(11.16–86.73) | **20.73(6.56–65.54)** |
| | Grade 1–4 completed | 32(28.3) | 58(15.2) | 14.56(5.4–39.27) | **7.62(2,58–22.51)** |
| | Grade 5–8 completed | 43(38.1) | 164(42.9) | 6.92(2.67–17.97) | **3.18(1.11–9.06)** |
| | Grade 9–10+ completed | 5(4.4) | 132(34.6) | 1.0 | 1.0 |
| Maternal Occupation | Housewife | 89(80.2) | 292(77.5) | 1.0 | 1.0 |
| | Employed/ | 22(19.8) | 85(22.5) | 0.54(0.50–1.44) | **2.79(1.41–5.52)** |
| Father's education | No class attended | 25(22.1) | 18(4.7) | 19.19(8.79–41.91) | |
| | Grade 1–4 completed | 15(13.3) | 21(5.5) | 9.87(4.32–22.54) | * |
| | Grade 5–8 completed | 56(49.6) | 108(28.3) | 7.17(3.98–12.91) | |
| | Grade 9–10+ completed | 17(15.0) | 235(61.5) | 1.0 | |
| Distance of house from health center | <30 minutes' walk | 36(32) | 177(46.3) | 1.0 | |
| | 30–60 minutes' walk | 77(68) | 205(53.7) | 1.85(1.18–2.88) | * |
| Total number of Family members | <5 members | 52(46) | 222(58) | 1.0 | |
| | ≥5members | 61(54) | 160(42) | 1.63(1.10–2.48) | * |
| No of Antenatal care visits (n = 474) | 1 | 4(4.2) | 6(1.6) | 4.42(1.18–16.51) | |
| | 2 | 26(27) | 49(13) | 3.51(1.92–6.43) | * |
| | 3 | 34(35.4) | 111(29.4) | 2.03(1.19–3.46) | |
| | 4+ | 32(33.3) | 212(56.2) | 1.0 | |
| Knew the due date for the index child | Yes | 59(52.2) | 288(75.4) | 1.0 | 1.00 |
| | No | 54(47.8) | 94(24.6) | 2.80(1.81–4.34) | **1.81 (1.03–3.18)** |
| Planned place of delivery for the index child | Yes | 101(89.4) | 381(99.7) | 1.0 | 1.0 |
| | No | 12(10.6) | 1(0.3) | 45.26(5.82–352.25) | **26.27(2.59–266.89)** |
| Birth order of the child | First order | 23(20.4) | 146(38.2) | 1.0 | |
| | 2-3rd order | 53(46.9) | 171(44.8) | 1.97(1.15–3.37) | * |
| | 4th or more order | 37(32.7) | 65(17.0) | 3.61(1.99–6.56) | |

figures reported by a study conducted in Bench Maji Zone (21.7%), Southern Ethiopia [19] and EDHS report of 2016 (25.3%) in Ethiopia [20]. However, the prevalence in our study is lower than that reported by studies in different parts of Ethiopia [21–24]. This difference might be attributed to various factors including differences in infrastructures such as roads and transport systems, which are critical to ensure access to healthcare facilities [25].

A relatively high prevalence of home delivery was observed in Shebedino district which is likely related to socio-cultural factors that need to be further explored. Overall, access to local mass media in local languages, access to health infrastructure and leadership, promotion efforts for institutional delivery and factors related to demographic and socioeconomic characteristics could affect preference of place of delivery of women.

More than two-thirds of respondents who participated in the survey were 21–30 years old implying the high age-specific fertility rate. One in ten women did not attend any formal education indicating that most women in the study area attended some education compared to figures reported by similar surveys [16, 17]. The better educational status could have attributed to the observed remarkable decline of home delivery in the study area compared to figures reported by EDHS 2016 [3].

Our findings showed that women residing in rural areas were five times more likely to give birth at home compared to those living in urban areas. Several similar studies reported that place of residence and distance from health facilities as common predictors of home birth. This could be related to the limited access to health information and service as well as long distance from the health facilities among rural dwellers limiting the chances of institutional birth [4, 17, 22, 26, 27]. Our study calls for better strategies to reach the rural community to avert home births.

Our findings show that level of maternal education and occupation predicted place of delivery. Uneducated women are less empowered and are unable to make use of multiple sources of information related to complicated pregnancies [15]. Moreover, uneducated women are also more likely to be influenced by socio-cultural phenomena which discourage institutional delivery. This was well demonstrated by this study and consistently supported by similar surveys [15, 22, 26]. Lack of knowledge on the due date demonstrated similar effect as maternal education. Women who did not know expected date of delivery are more likely to deliver at home compared to those who know it. Mothers who forget expected date of delivery are less likely to get prepared and plan for institutional delivery.

One unique observation of the current study is that employed women were found to be more likely to deliver at home compared to housewives. Employed women are considered to be more mobile because of the nature of their work, more likely to use contraceptives and have better awareness compared to housewives. The finding of this study is contrary to this reality requiring further investigation. Lack of early plan of place of delivery and lack of preparedness for institutional delivery due to their busy daily routine are among the factors that might increase the probability that the employed mother could give birth at home.

## Limitation of the study

All determinant factors couldn't be assessed by this questionaries-based survey as additional qualitative component could help in ascertaining them. However, this survey is the first of its type for the site to reveal reasons for home births and assists policy makers and health system leadership to design ways of tackling the issues. In this study, numbers of ANC visits, distance of health facility from home, and total number of family size were not associated with home delivery mainly because of inadequate power of the test. The 95%CI for some aORs also indicate the effect of small sample size on the precision of estimated measures of reported effect size.

## Conclusions and recommendations

The prevalence of home birth has significantly dropped in the study setting compared to figures reported in 2016. Parental and maternal education, mother's occupation, and place of residence and planned place of delivery were found to be predictors of home delivery. We recommend further study using qualitative methods and interventions targeting rural and uneducated parents to further reduce home births.

## Supporting information

**S1 Data.**
(SAV)

**S2 Data.**
(BIN)

**S1 Questionnaire.**
(DOC)

## Acknowledgments

We thank Sidama Administration Health Department, district health offices and kebele administrations in the study area for their support. We extend our thanks to data collectors and respondents for their time and willingness to participate in the study.

## Author Contributions

**Conceptualization:** Dejene Hailu, Henok Tadele, Birkneh Tilahun Tadesse, Selamawit Mengesha.

**Data curation:** Yusuf Haji.

**Formal analysis:** Dejene Hailu, Birkneh Tilahun Tadesse, Akalewold Alemayehu, Yusuf Haji.

**Funding acquisition:** Henok Tadele.

**Investigation:** Henok Tadele, Teshome Abuka, Abel Gedefaw, Yusuf Haji.

**Methodology:** Dejene Hailu, Akalewold Alemayehu, Abel Gedefaw.

**Project administration:** Dejene Hailu, Teshome Abuka.

**Supervision:** Dejene Hailu, Fitsum Woldegebriel, Selamawit Mengesha.

**Validation:** Selamawit Mengesha.

**Visualization:** Fitsum Woldegebriel.

**Writing – original draft:** Dejene Hailu.

**Writing – review & editing:** Henok Tadele, Birkneh Tilahun Tadesse, Fitsum Woldegebriel.

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
