## [Decision Letter · Decision Letter 0]

15 Feb 2021

PONE-D-20-23821

Institutional delivery practice in southern Ethiopia.

PLOS ONE

Dear Dr. Kassa,

Thank you for submitting your manuscript to PLOS ONE. After careful consideration, we feel that it has merit but does not fully meet PLOS ONE’s publication criteria as it currently stands. Therefore, we invite you to submit a revised version of the manuscript that addresses the points raised during the review process.

Thank you for submitting your manuscript to PLOS ONE. After careful consideration, we feel that it has merit but does not fully meet PLOS ONE’s publication criteria as it currently stands. Therefore, we invite you to submit a revised version of the manuscript that addresses the points raised during the review process.

The main limitation of the current study is that it can be seen as a replication study of the numerous similar studies carried out in different regions of the country. Many of them are reported, some of the more recent ones are missing. PLOS ONE policy regarding replication studies is that if a submitted study replicates or is very similar to previous work, authors must provide a sound scientific rationale for the submitted work and clearly reference and discuss the existing literature. Submissions that replicate or are derivative of existing work will likely be rejected if authors do not provide adequate justification. The authors should streghthen the rationale for the submitted work and ensure they clearly reference the existing literature.

Regarding some of the key missing references: https://doi.org/10.1186/s12889-019-7854-2 is interesting since it works with the DHS survey mentioned in the introduction, DOI: 10.1186/s12884-017-1409-2 looks also at the relationship with seeking antenatal care..., DOI: 10.1186/s12889-020-08919-8 ... I believe there are more references missing.

Another limitation of this study is sample size and low power. This means that large differences in relative risk such as those of antenatal care, seem not be significant. I say seem, because there are problems in reporting. It is said that a screening based on p-values of 0.2 is carried out in a first stage, but those p-values are missing from table 3 (or elsewhere), they should be included.

There is also no treatment of multicolllinearity. Several of the variables are probably closely related, such as urban/rural and distance to health facility. This might explain changes of signs. Some evaluation of multicollinearity is needed. I suggest one of two strategies: the first one, you could use a backward/forward model selection procedure similar to the Debremarkos study mentioned above. Otherwise, you could report the R2 of the covariiates with respect to the rest of covariates (VIF, equal to 1/1-R2aux is better avoided since it does not provide a variance inflation factor in the context of logistic regression).

Due to the large number of categorical variables and small sample size, there could be problems of complete or quasi-complete separation. Please report on the convergence of estimates (at least in the review report) since lack of convergence is often due to problems of separation.

In addition, PLOS ONE policy is that the data is provided before publication. Please follow PLOS ONE policies including the data in the submission or providing a link to an open repository where the data can be accessed.

Look also at the additional points made by the reviewers.

We look forward to receiving your revised manuscript.

Kind regards,

José Antonio Ortega, Ph.D.

Academic Editor

PLOS ONE

Journal Requirements:

2. In the Methods, please clarify that participants provided oral consent. Please also state in the Methods:

- Why written consent could not be obtained

- Whether the Institutional Review Board (IRB) approved use of oral consent

- How oral consent was documented

For more information, please see our guidelines for human subjects research: https://journals.plos.org/plosone/s/submission-guidelines#loc-human-subjects-research

3. You indicated that you had ethical approval for your study. In your Methods section, please clarify a) if any minors were included in your study and if so, b) ensure you have also stated whether you obtained consent from parents or guardians of the minors included in the study or whether the research ethics committee or IRB specifically waived the need for their consent.

4. In your Methods section, please provide additional information about the participant recruitment method and the demographic details of your participants. Please ensure you have provided sufficient details to replicate the analyses such as: a) a description of any inclusion/exclusion criteria that were applied to participant recruitment, b) a statement as to whether your sample can be considered representative of a larger population.

5.We note that you have indicated that data from this study are available upon request. PLOS only allows data to be available upon request if there are legal or ethical restrictions on sharing data publicly. For information on unacceptable data access restrictions, please see http://journals.plos.org/plosone/s/data-availability#loc-unacceptable-data-access-restrictions.

6.We note that the grant information you provided in the ‘Funding Information’ and ‘Financial Disclosure’ sections do not match.

7. PLOS requires an ORCID iD for the corresponding author in Editorial Manager on papers submitted after December 6th, 2016. Please ensure that you have an ORCID iD and that it is validated in Editorial Manager. To do this, go to ‘Update my Information’ (in the upper left-hand corner of the main menu), and click on the Fetch/Validate link next to the ORCID field. This will take you to the ORCID site and allow you to create a new iD or authenticate a pre-existing iD in Editorial Manager. Please see the following video for instructions on linking an ORCID iD to your Editorial Manager account: https://www.youtube.com/watch?v=_xcclfuvtxQ

8. We note that you have included the phrase “data not shown” in your manuscript. Unfortunately, this does not meet our data sharing requirements. PLOS does not permit references to inaccessible data. We require that authors provide all relevant data within the paper, Supporting Information files, or in an acceptable, public repository. Please add a citation to support this phrase or upload the data that corresponds with these findings to a stable repository (such as Figshare or Dryad) and provide and URLs, DOIs, or accession numbers that may be used to access these data. Or, if the data are not a core part of the research being presented in your study, we ask that you remove the phrase that refers to these data.

9. Please amend either the abstract on the online submission form (via Edit Submission) or the abstract in the manuscript so that they are identical.

10. Please include captions for your Supporting Information files at the end of your manuscript, and update any in-text citations to match accordingly. Please see our Supporting Information guidelines for more information: http://journals.plos.org/plosone/s/supporting-information.

11. Your ethics statement should only appear in the Methods section of your manuscript. If your ethics statement is written in any section besides the Methods, please delete it from any other section.

Reviewers' comments:

Reviewer's Responses to Questions

**Comments to the Author**

1. Is the manuscript technically sound, and do the data support the conclusions?

Reviewer #1: Yes

Reviewer #2: Yes

Reviewer #3: Yes

2. Has the statistical analysis been performed appropriately and rigorously? 

Reviewer #1: Yes

Reviewer #2: Yes

Reviewer #3: Yes

3. Have the authors made all data underlying the findings in their manuscript fully available?

Reviewer #1: Yes

Reviewer #2: Yes

Reviewer #3: Yes

4. Is the manuscript presented in an intelligible fashion and written in standard English?

Reviewer #1: Yes

Reviewer #2: Yes

Reviewer #3: Yes

5. Review Comments to the Author

Reviewer #1: How could you see the significance of the study as many studies published in Ethiopia including DHS?. (may be as requirements for KMC project)As a report made by authors the study sampling procedure is multistage and design effect of 2 was used to calculate sample size but only three districts selected to represent the whole 36districts in Sidaama region mentioned in the title. 

How do you see generalization to the whole region?

and also why you are interested in urban and rural division in spite of the fact that statistical difference and separate factors for both was not appreciated(32 versus 3.2 Percent)

Need further discussion for maternal merchant occupation as risk factors for home delivery since women economic empowerment pave the way for institutional delivery?

the issues of wide confidence interval for prominent factors called planned place of delivery and chi square assumption which is not fulfilled(1)

The limitations didn't go with study objectives? Since the trend is not many objectives of current study?

Reviewer #2: Very useful manuscript. Major revisions are required as the discussion is repetition of results. Discussion needs to be more of explanation for findings and comparison with other studies. Need to be rewritten.

Authors do not compare data across districts. No description of graph of district wise deliveries in the results but mentioned in the discussion but no elaboration.

Discrepancy of merchant women having more home births needs to be discussed in a detail.

Reviewer #3: 1. Title says "Institutional delivery practice in southern Ethiopia" but the aim is to assess prevalence and determinants of of home delvery. Better to change it " Home delivery practice and its predictors in South Ethiopia"

2. Discussion part: first paragraph should only summarize the main findings, the interpretation and comparisons should come in the subsquent paragrpahs.

3. In the limitation part: it is stated " “Being cross sectional, this study may not give picture on the trends of delivery” but the aim of your study is to determine prevalence of home delivery and factors associated, and you answered that, I don’t think the cross-sectional design is a limitation here.

6. PLOS authors have the option to publish the peer review history of their article (what does this mean?). If published, this will include your full peer review and any attached files.

Reviewer #1: No

Reviewer #2: **Yes: **Kranti Suresh Vora

Reviewer #3: No

---

## [Author Response · Author response to Decision Letter 0]

10 Jun 2021

Point by point response letter 

Dear Reviewers and Editorial Team,

Thank you for taking your time to review our manuscript entitled ‘institutional delivery practice in Southern Ethiopia’ and for forwarding suggestions to help improve our manuscript. 

Kindly find below the responses given to the comments and suggestions. 

Responses to the most recent questions forwarded on April 29, 2021

Thank you for submitting your manuscript entitled "Home Delivery Practice and its Predictors in South Ethiopia" to PLOS ONE. Your manuscript files have been checked in-house but before we can proceed we need you to address the following issues:

1. Thank you for your recent responses. Thank you for clarifying the verbal consent procedure. Please update the manuscript and the online submission form to clarify how verbal consent was documented.

Response: The information sheet and consent was read slowly and loudly by the data collector to the participants. Then, they were asked if there were any queries. After the mothers had confirmed that all is clear, they were asked one last question if they were willing to participate in the survey or not. The data collectors circled either ‘yes’ or ‘no’ based on whichever is selected and the interview was conducted only if the data collector was told to circle the response ‘yes’. This was attached to the questionnaire and documented. For more details, please, refer to the section ‘Ethical approval and consent to participate’, in the manuscript on page 6)

Responses to the questions forwarded on April 1, 2021:

1. Please amend the title either on the online submission form or in your manuscript so that they are identical.

Response: based on the suggestion forwarded, the title “Institutional Delivery Practice in Southern Ethiopia” was amended as “Home Delivery Practice and its Predictors in South Ethiopia”

2) Thank you for clarifying the verbal consent procedure. Please update the manuscript and the online submission form to clarify how verbal consent was documented.

Response: Done as per the suggestion

3) Your demographic table indicates that 18.6% of your participants were under the age of 20. Please clarify whether or not your study involved minors (under the age of 18) and if so, whether or not consent was sought from the participants' parents or guardians. If the IRB specifically waived the requirement for parental/guardian consent, please include that information.

Response: Seven (1.4%) of women involved in this survey were less than 18 years old. However, since all women participated in the survey was married, they live with their husbands not with the parents and as a result, the consent was not sought from the parents which was waived by the IRB considering the early marriage practices (, 18 year of age) in Ethiopia. 

3. Thank you for uploading your study's underlying data set, (home Delivery survey.sav). We noticed that this file may contain potentially identifying participant information (Patient ID's in column 1).

 Response: Potential identifiers such as house number, individual ID numbers and name of the villages (kebeles) have been removed from the current data set.

The following include responses to the general and specific comments raised by the editor and reviewers which were sent in February, 2021. 

1. General comments

Comment 1.1: The main limitation of the current study is that it can be seen as a replication study of the numerous similar studies carried out in different regions of the country. Many of them are reported, some of the more recent ones are missing. PLOS ONE policy regarding replication studies is that if a submitted study replicates or is very similar to previous work, authors must provide a sound scientific rationale for the submitted work and clearly reference and discuss the existing literature. Submissions that replicate or are derivative of existing work will likely be rejected if authors do not provide adequate justification. The authors should strengthen the rationale for the submitted work and ensure they clearly reference the existing literature.

Response 1.1: Our study was undertaken in an area where no similar studies had been conducted and this is not a replicated study. The other point is that when we had implemented a kangaroo mother care service in Sidama Zone preceding the current study, the local health authorities were unable to tell us the exact proportion of home delivery so that we could plan to reach low-birth weight babies born at home who could be eligible for KMC service. Hence, this finding informs the local health authorities the best estimate of home delivery in their zone.

Comment1.2: Regarding some of the key missing references: https://doi.org/10.1186/s12889-019-7854-2 is interesting since it works with the DHS survey mentioned in the introduction, DOI: 10.1186/s12884-017-1409-2 looks also at the relationship with seeking antenatal care..., DOI: 10.1186/s12889-020-08919-8 ... I believe there are more references missing.

Response 1.2: Now we have included the additional references forwarded to us. 

Comment 1.3: Another limitation of this study is sample size and low power. This means that large differences in relative risk such as those of antenatal care, seem not be significant. I say seem, because there are problems in reporting. It is said that a screening based on p-values of 0.2 is carried out in a first stage, but those p-values are missing from table 3 (or elsewhere), they should be included.

Response 1.3: We appreciate the comment. We have discussed this under the section ‘Limitation of the study’. A p-value<0.2 was used for crude analysis to screen independent variables for the final mode (adjusted analysis). We omitted inclusion of this p-value for two reasons. The first is that it is part of preliminary steps we have gone through in data analysis and is not part of the main findings to be discussed. The second is we wanted to reduce the number of columns in table 3 and we believe that the 95%CI reported in the crude analysis could be more informative. 

Comment 1.4: There is also no treatment of multicolllinearity. Several of the variables are probably closely related, such as urban/rural and distance to health facility. This might explain changes of signs. Some evaluation of multicollinearity is needed. I suggest one of two strategies: the first one, you could use a backward/forward model selection procedure similar to the Debremarkos study mentioned above. Otherwise, you could report the R2 of the covariates with respect to the rest of covariates (VIF, equal to 1/1-R2aux is better avoided since it does not provide a variance inflation factor in the context of logistic regression).

Response 1.4: Based on the suggestion, a backward stepwise logistic regression model was run and accordingly, outputs presented in table 3 were revised.

Comment 1.5: Due to the large number of categorical variables and small sample size, there could be problems of complete or quasi-complete separation. Please report on the convergence of estimates (at least in the review report) since lack of convergence is often due to problems of separation.

In addition, PLOS ONE policy is that the data is provided before publication. Please follow PLOS ONE policies including the data in the submission or providing a link to an open repository where the data can be accessed.

Response 1.5: Thank you for raising this important point which is a likely problem in logistic regression. Remedial action such as increasing sample size to overcome such problem is not practical at this stage. However, during analysis, we have observed no sign of complete or quasi-complete separation reported by the model in the output. We have now uploaded the data. 

Comment 1.6: Please ensure that your manuscript meets PLOS ONE's style requirements, including those for file naming. The PLOS ONE style templates can be found at

 Response1.6: Checked and corrected. Thanks!

Comment 2: In the Methods, please clarify that participants provided oral consent. Please also state in the Methods:

- Why written consent could not be obtained

- Whether the Institutional Review Board (IRB) approved use of oral consent

- How oral consent was documented

For more information, please see our guidelines for human subjects research: https://journals.plos.org/plosone/s/submission-guidelines#loc-human-subjects-research

 Response 2: Now the consent steps are expanded in the manuscript based on the comments. Please, see the section ‘Ethical approval and consent to participate’ of the revised MS. 

Comment 3: You indicated that you had ethical approval for your study. In your Methods section, please clarify a) if any minors were included in your study and if so, b) ensure you have also stated whether you obtained consent from parents or guardians of the minors included in the study or whether the research ethics committee or IRB specifically waived the need for their consent.

Response 3: Our study didn’t involve minors. Our study subjects were mothers who gave birth in the last 12months prior to the survey. The seven (1.4%) mothers less than 18 years old who were participated in this survey were already married and do not live with the parents. Verbal consent to participate in the study was obtained from these mothers and approved by the IRB.

 Comment 4: In your Methods section, please provide additional information about the participant recruitment method and the demographic details of your participants. Please ensure you have provided sufficient details to replicate the analyses such as: a) a description of any inclusion/exclusion criteria that were applied to participant recruitment, b) a statement as to whether your sample can be considered representative of a larger population.

 Response 4: Additional points are included on exclusion criteria based on comments. Concerning representation of a large population, we calculated sample based on the national demographic health survey proportion with inclusion of design effect. Hence, our study included representative population with inclusion of rural and urban settings making generalizability and representation more acceptable. 

Comment 5: We note that you have indicated that data from this study are available upon request. PLOS only allows data to be available upon request if there are legal or ethical restrictions on sharing data publicly. For information on unacceptable data access restrictions, please see http://journals.plos.org/plosone/s/data-availability#loc-unacceptable-data-access-restrictions.

 Response 5: Based on the comment, we have uploaded the data analyzed for this study and there is no restriction imposed by any one.

Comment 6: In your revised cover letter, please address the following prompts:

 Response 6: There is no restriction imposed on the data (uploaded). 

 Comment 7: We note that the grant information you provided in the ‘Funding Information’ and ‘Financial Disclosure’ sections do not match. When you resubmit, please ensure that you provide the correct grant numbers for the awards you received for your study in the ‘Funding Information’ section

 Response 7: Noted and corrected. Funding was obtained from Gates Foundation through the WHO

Comment 8: PLOS requires an ORCID iD for the corresponding author in Editorial Manager on papers submitted after December 6th, 2016. Please ensure that you have an ORCID iD and that it is validated in Editorial Manager. To do this, go to ‘Update my Information’ (in the upper left-hand corner of the main menu), and click on the Fetch/Validate link next to the ORCID field. This will take you to the ORCID site and allow you to create a new iD or authenticate a pre-existing iD in Editorial Manager. Please see the following video for instructions on linking an ORCID iD to your Editorial Manager account: https://www.youtube.com/watch?v=_xcclfuvtxQ

Response 8: Noted and done 

Comment 9: We note that you have included the phrase “data not shown” in your manuscript. Unfortunately, this does not meet our data sharing requirements. PLOS does not permit references to inaccessible data. We require that authors provide all relevant data within the paper, Supporting Information files, or in an acceptable, public repository. Please add a citation to support this phrase or upload the data that corresponds with these findings to a stable repository (such as Figshare or Dryad) and provide and URLs, DOIs, or accession numbers that may be used to access these data. Or, if the data are not a core part of the research being presented in your study, we ask that you remove the phrase that refers to these data.

 Response 9: The sentence is removed as suggested. 

Comment 10: Please amend either the abstract on the online submission form (via Edit Submission) or the abstract in the manuscript so that they are identical.

 Response 10: Done as suggested

Comment 11: Please include captions for your Supporting Information files at the end of your manuscript, and update any in-text citations to match accordingly. Please see our Supporting Information guidelines for more information: http://journals.plos.org/plosone/s/supporting-information.

 Response: done as suggested

Comment 12: Your ethics statement should only appear in the Methods section of your manuscript. If your ethics statement is written in any section besides the Methods, please delete it from any other section.

 Response 12: It is corrected. 

Review Comments to the Author 

Reviewer #1: How could you see the significance of the study as many studies published in Ethiopia including DHS? (may be as requirements for KMC project) As a report made by authors the study sampling procedure is multistage and design effect of 2 was used to calculate sample size but only three districts selected to represent the whole 36districts in Sidaama region mentioned in the title. 

How do you see generalization to the whole region? and also why you are interested in urban and rural division in spite of the fact that statistical difference and separate factors for both was not appreciated(32 versus 3.2 Percent)

Need further discussion for maternal merchant occupation as risk factors for home delivery since women economic empowerment pave the way for institutional delivery? The issues of wide confidence interval for prominent factors called planned place of delivery and chi square assumption which is not fulfilled(1)The limitations didn't go with study objectives? Since the trend is not many objectives of current study?

Response to comments of Reviewer 1

- Representation to the wider Sidama Region from this survey: We have calculated sample size based on the DHS figure reported in 2016. The sampling was done from the districts where the KMC implementation research was undertaken. As we have included rural and urban settings quite similar to other districts, we feel the generalizability to the whole Sidama region is acceptable. 

- Concerning the urban and rural setting interesting despite absence of statistical difference: We used urban and rural settings during sample size calculation for allocation of sample to each. There is statistical difference between urban and rural settings as to home delivery, mothers residing in the rural area have five times higher odd of home birth (Table 3). 

- Merchant occupation and home delivery: As we described it in the discussion part, it is contrary to the expectation. We have highlighted that it could be related to their busy daily routine and missing scheduled ANC appointments to plan the delivery ahead. 

Reviewer #2: Very useful manuscript. Major revisions are required as the discussion is repetition of results. Discussion needs to be more of explanation for findings and comparison with other studies. Need to be rewritten.

Response: Corrected. 

Authors do not compare data across districts. No description of graph of district wise deliveries in the results but mentioned in the discussion but no elaboration.

Response: It was shown in figure 1 under the section ‘characteristics of deliveries’. This point was discussed in the second paragraph of the discussion. 

Discrepancy of merchant women having more home births needs to be discussed in a detail.

Response: We have added few points on the discussion part. It is subject to further study. 

Reviewer #3: 

1. Title says "Institutional delivery practice in southern Ethiopia" but the aim is to assess prevalence and determinants of home delivery. Better to change it " Home delivery practice and its predictors in South Ethiopia"

Response 1: Comment is accepted and corrected

2. Discussion part: first paragraph should only summarize the main findings; the interpretation and comparisons should come in the subsequent paragraphs.

Response 2: Comment accepted and corrected. 

3. In the limitation part: it is stated " “Being cross sectional, this study may not give picture on the trends of delivery” but the aim of your study is to determine prevalence of home delivery and factors associated, and you answered that, I don’t think the cross-sectional design is a limitation here.

Response 3: Comment is accepted and correction is made. 

 I thank all the reviewers and the editorial board members for their important comments and questions provided 

Sincerely yours,

Dejene Hailu Kassa (PhD)

---

## [Decision Letter · Decision Letter 1]

2 Jul 2021

Home Delivery Practice and its Predictors in South Ethiopia

PONE-D-20-23821R1

Dear Dr. Kassa,

We’re pleased to inform you that your manuscript has been judged scientifically suitable for publication and will be formally accepted for publication once it meets all outstanding technical requirements.

Kind regards,

José Antonio Ortega, Ph.D.

Academic Editor

PLOS ONE

Additional Editor Comments (optional):

There are some editing suggestions by reviewer 3 that should be incorporated.

Also I have noticed that in the main text you refer to merchant women when the category includes all employed women (including self-employed). I would suggest changing merchant to employed women throughout, making sure the first time to mention that employed women include those working for wage and self-employed.

Regarding the backwards-selection GLM regression, it is important to specify in the methods section the procedure employed (alpha level chosen for removing, or AIC maximization, ...).. Also note that PLOS ONE criteria for statistical reporting ask you to Include the full results of any regression analysis performed as a supplementary file. Include all estimated regression coefficients, their standard error, p-values, and confidence intervals, as well as the measures of goodness of fit. I'd suggest including the first full model estimated and the final model selected by the backward selection algorithm.

Reviewers' comments:

Reviewer's Responses to Questions

**Comments to the Author**

1. If the authors have adequately addressed your comments raised in a previous round of review and you feel that this manuscript is now acceptable for publication, you may indicate that here to bypass the “Comments to the Author” section, enter your conflict of interest statement in the “Confidential to Editor” section, and submit your "Accept" recommendation.

Reviewer #2: All comments have been addressed

Reviewer #3: (No Response)

2. Is the manuscript technically sound, and do the data support the conclusions?

Reviewer #2: Yes

Reviewer #3: Yes

3. Has the statistical analysis been performed appropriately and rigorously? 

Reviewer #2: Yes

Reviewer #3: Yes

4. Have the authors made all data underlying the findings in their manuscript fully available?

Reviewer #2: Yes

Reviewer #3: Yes

5. Is the manuscript presented in an intelligible fashion and written in standard English?

Reviewer #2: Yes

Reviewer #3: Yes

6. Review Comments to the Author

Reviewer #2: (No Response)

Reviewer #3: 1. Title is corrected as suggested in the previous review report; which is good.

2. There are few editorial correction which need to be corrected to show some of them

-Introduction last paragraph, last sentence, better be corrected to read " to assess the prevalence of homebirth and determinants, omit "rate"

-Results: sociodemograph characterstics: last sentence "377 (76.2%) of the women were housewives and 45% of had at least'' either remove "of" or add "them" after it.

- Under the heading Antenatal care follow up and plan for place of deliverey: first paragraph , last sentence "preferred their women,relatives" I think you need to omit the coma betwen women and relative.

- Conclusion and recommendation: don't you think it is important to recommend further study using qualitative methods, as it allows to get depth information about the issue. There are recent quaitative studies from Ethiopia on similar topics.

7. PLOS authors have the option to publish the peer review history of their article (what does this mean?). If published, this will include your full peer review and any attached files.

Reviewer #2: **Yes: **Kranti Suresh Vora

Reviewer #3: No

---

## [Editor Report · Acceptance letter]

30 Jul 2021

PONE-D-20-23821R1 

Home Delivery Practice and its Predictors in South Ethiopia 

Dear Dr. Hailu:

I'm pleased to inform you that your manuscript has been deemed suitable for publication in PLOS ONE. Congratulations! Your manuscript is now with our production department. 

Kind regards, 

on behalf of

Dr. José Antonio Ortega 

Academic Editor

PLOS ONE